# The associations between functional dyspepsia and potential risk factors: A comprehensive Mendelian randomization study

**Zeyu Wang**◉, **Tangyi Liu**◉, **Dan Cao**◉, **Hui Luo**, **Ze Yang**, **Xiaoyu Kang**\*, **Yanglin Pan**◉\*

State Key Laboratory of Holistic Integrative Management of Gastrointestinal Cancers and National Clinical Research Center for Digestive Diseases Xijing Hospital of Digestive Diseases, Fourth Military Medical University, Xi an, China

◉ These authors contributed equally to this work.
\* yanglinpan@hotmail.com (YP); kangxiaoyu@hotmail.com (XK)

## Abstract

### Background

Previous cross-sectional studies have identified multiple potential risk factors for functional dyspepsia (FD). However, the causal associations between these factors and FD remain elusive. Here we aimed to fully examine the causal relationships between these factors and FD utilizing a two-sample MR framework.

### Methods

A total of 53 potential FD-related modifiable factors, including those associated with hormones, metabolism, disease, medication, sociology, psychology, lifestyle and others were obtained through a comprehensive literature review. Independent genetic variants closely linked to these factors were screened as instrumental variables from genome-wide association studies (GWASs). A total of 8875 FD cases and 320387 controls were available for the analysis. The inverse variance weighted (IVW) method was employed as the primary analytical approach to assess the relationship between genetic variants of risk factors and the FD risk. Sensitivity analyses were performed to evaluate the consistency of the findings using the weighted median model, MR-Egger and MR-PRESSO methods.

### Results

Genetically predicted depression (OR 1.515, 95% confidence interval (CI) 1.231 to 1.865, p = 0.000088), gastroesophageal reflux disease (OR 1.320, 95%CI 1.153 to 1.511, p = 0.000057) and years of education (OR 0.926, 95%CI 0.894 to 0.958, p = 0.00001) were associated with risk for FD in univariate MR analyses. Multiple medications, alcohol consumption, poultry intake, bipolar disorder, mood swings, type 1 diabetes, elevated systolic blood pressure and lower overall health rating showed to be suggestive risk factors for FD

**Data Availability Statement:** All data relevant to the study are included in the article or uploaded as supplementary information. (1) Psychiatric Genomics Consortium (PGC) (https://pgc.unc.edu/

); (2) GWAS and Sequencing Consortium of Alcohol and Nicotine use (GSCAN) (https://conservancy.umn.edu/handle/11299/201564); (3) Genetic Investigation of Anthropometric Traits Consortium (GIANT) (http://portals.broadinstitute.org/collaboration/giant/); (4) Meta-analyses of Glucose and Insulin-related traits Consortium (MAGIC) (https://magicinvestigators.org/downloads/); (5) MRC Integrative Epidemiology Unit (MRC-IEU); (6) Neale Lab (http://www.nealelab.is/uk-biobank). The summary statistics are available on the public platforms.

**Funding:** This work supported in part by National Key R&D Program of China (2022YFC2505100 to PYL) and the National Natural Science Foundation of China (81970557 to PYL and 82373117 to KXY). The funders had no role in study design, data collection and analysis, decision to publish, or preparation of the manuscript. There are no any commercial or other interests or associations that might be perceived as posing a conflict of interest or bias in connection with the submitted article.

**Competing interests:** The authors have declared that no competing interests exist.

**Abbreviations:** FD, Functional dyspepsia; GWAS, Genome-wide association study; IVW, Inverse variance weighted; MR, Mendelian randomization; WM, weighted median; MR-PRESSO, the MR pleiotropy residual sum and outlier; MVMR, Multivariable Mendelian randomization; CI, Confidence Interval; SNPs, single nucleotide polymorphisms; IBS, Irritable bowel syndrome; GERD, Gastroesophageal reflux disease; ADHD, Attention Deficit Hyperactivity Disorder; BMI, Body mass index; NAFLD, Nonalcoholic fatty liver disease; GIANT, Genetic Investigation of ANthropometric Traits; GSCAN, GWAS and Sequencing Consortium of Alcohol and Nicotine use; MAGIC, Meta-Analyses of Glucose and Insulin-related traits Consortium; PGC, Psychiatric Genomics Consortium; MRC-IEU, MRC Integrative Epidemiology Unit; GSCAN, GWAS and Sequencing Consortium of Alcohol and Nicotine use.

(all p<0.05 while ≥0.00167). The positive causal relationship between depression, years of education and FD was still significant in multivariate MR analyses.

## Conclusions

Our comprehensive MR study demonstrated that depression and lower educational attainment were causal factors for FD at the genetic level.

## Introduction

Functional dyspepsia (FD) is a complex disorder featured with chronic and recurrent upper abdominal discomfort or pain, which cannot be explained by biochemical or structural abnormalities [1]. The pooled global prevalence of FD ranged from 5% to 20% [2, 3]. Despite of normal life expectancy, the impact of FD on patients's quality of life and social function is substantial. Several studies reported that FD was associated with reduced life quality [4], lower work efficiency [5] and increased medical costs [6]. Given the unsatisfactory efficiency of current therapeutic approaches [7], it is important to identify risk factors for FD. The early identification and targeted intervention would contribute to alleviating those harmful consequences.

Several modifiable risk factors, including lifestyle factors (e.g. smoking and heavy intake of chili [3, 8]), psychological factors (e.g. anxiety and depression [9–11]), medication use factors (e.g. use of non-steroidal anti-inflammatory drugs (NSAIDs) [12]), disease-related factors (e.g. H pylori infection [3, 13]), metabolism-related factors (e.g. BMI [13]) and sociological factors (e.g. education level [14]) were previously reported to be highly associated with FD in observational studies. However, those conventional studies failed to cover all potential risk factors of FD. In addition, measuring the causal impacts of modifiable factors on FD could be quite a challenge in observational studies, for the existence of underlying confounding or reverse causality may mislead the associations. Thus, It is crucial to determine whether these modifiable factors act causal roles in the development of FD or just serve as confounder profiles in a comprehensive method.

MR is an emerging method that is rapidly applied to explore the causal relationship between risk factors and clinical consequences utilizing genetic variants as instrumental variables. It is less likely to encounter confounding bias and reverse causality errors [15]. MR analyses have been successfully applied to identify the causal links between a variety of factors and many diseases [16, 17]. However, no study has been performed that uses MR analyses to explore the associations between modifiable risk factors and FD. The identification of the causal risk factors for FD would help to deliver more effective prevention strategies. Here, we conducted this study to examine the causal relationships between 53 genetically predominant factors and FD utilizing a two-sample MR framework.

## Methods

### MR design

This MR study is based on the following three assumptions: (1) genetic variants are closely linked to risk factors; (2) genetic variants are irrelevant to confounding factors; and (3) genetic variants influence outcomes merely through risk factors (Fig 1). To fully screen all relevant risk factors, we conducted an exhaustive search for possible risk factors associated with FD in PUBMED until Sep 23, 2023. The keywords were presented as follows: ("Risk Factors" [All

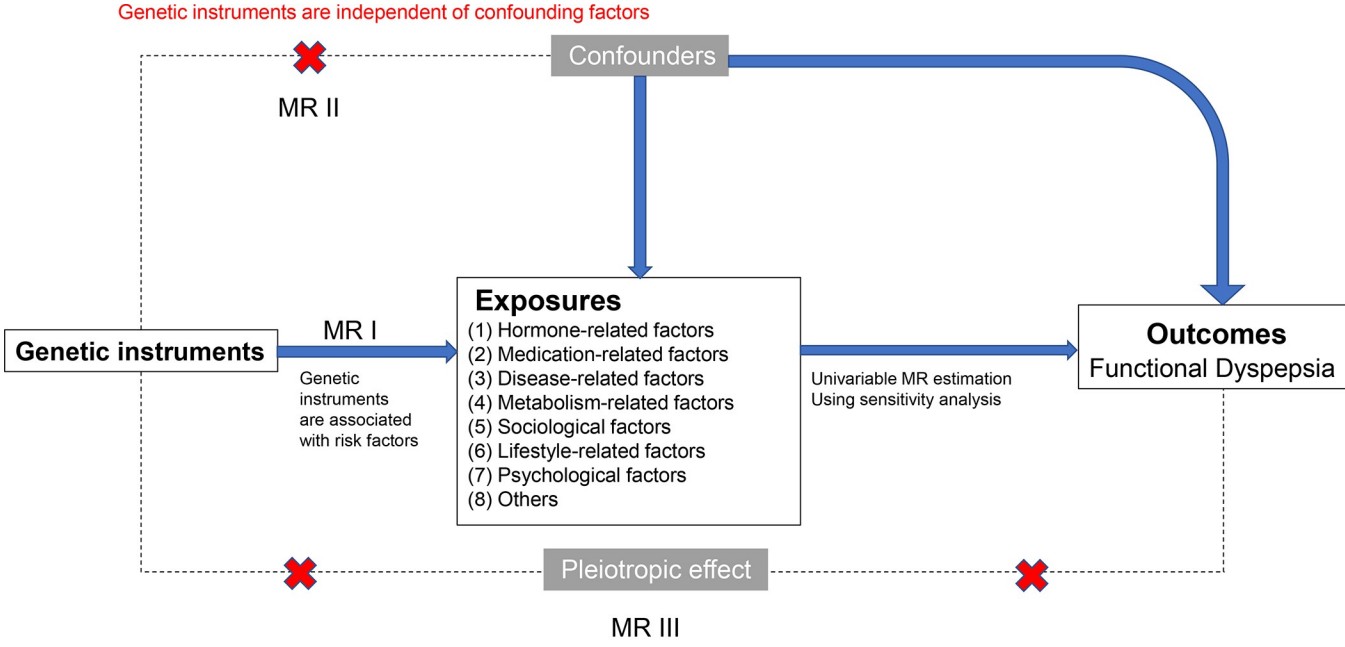

**Fig 1. Overview of the design and assumptions in this Mendelian randomization study.** Abbreviations: MR, Mendelian randomization.

Fields] OR "Risky" [All Fields] OR "Risk Factor" [All Fields] OR "Risk" [All Fields] OR "Factor" [All Fields] OR "Modifier" [All Fields] OR "Etiology" [All Fields] OR "pathogenesis"[All Fields] AND ("functional dyspepsia" [All Fields] OR "dyspepsia" [All Fields] OR "uninvestigated dyspepsia"[All Fields] OR "brain-gut interaction"[All Fields] OR "DBGI" [All Fields]). This search yielded a total of 5825 articles. We then refined the search to include only full-text articles or abstracts published after 1989, when Rome criteria was initially established. The selected categories encompassed meta-analyses, reviews, and systematic reviews, which subsequently reduced the number of relevant literatures to 954. The identified factors, excluding those without GWAS summary statistics such as Wheat and dietary fats, are listed S1 Table in S2 File. A sum of 53 modifiable risk factors were selected and categorized into eight subgroups, which were related to hormone, medication, disease, metabolism, sociology, lifestyle, psychology and others, respectively. The study was presented based on STORBE-MR guidelines (S1 File).

## Selection of genetic variants

We initially explored the relationships between possibly modifiable risk factors and FD using instrumental variables. The risk factors were extracted and classified into eight categories, as shown in Table 1. We extracted corresponding genetic instrumental variables from the following sources: (1) Psychiatric Genomics Consortium (PGC) (https://pgc.unc.edu/); (2) GWAS and Sequencing Consortium of Alcohol and Nicotine use (GSCAN) (https://conservancy.umn.edu/handle/11299/201564); (3) Genetic Investigation of Anthropometric Traits Consortium (GIANT) (http://portals.broadinstitute.org/collaboration/giant/); (4) Meta-analyses of Glucose and Insulin-related traits Consortium (MAGIC) (https://magicinvestigators.org/downloads/); (5) MRC Integrative Epidemiology Unit (MRC-IEU); (6) Neale Lab (http://www.nealelab.is/uk-biobank). The summary statistics are available on the public platforms.

**Table 1. Characteristics of the GWAS summary data.**

| Exposure | Ethnicity | Consortium | Total population | PMID | F- statistics | $R^2$ |
|---|---|---|---|---|---|---|
| **Hormone-related factors** | | | | | | |
| Testosterone levels | European | UK Biobank | 353805 | 34017140 | 79.7438 | 0.0208 |
| Estradiol levels | European | NA | 163985 | 34255042 | 38.3632 | 0.0005 |
| **Medication-related factors** | | | | | | |
| Medication code: antihypertensives | European | NA | 152380 | 34594039 | 38.7573 | 0.0018 |
| Medication code: antidepressants | European | NA | 304162 | 34594039 | 38.8816 | 0.0036 |
| Medication code: aspirin | European | MRC-IEU | 462933 | NA | 45.4663 | 0.0012 |
| Medication code: ibuprofen | European | MRC-IEU | 462933 | NA | 42.1607 | 0.0003 |
| Medication code: paracetamol | European | MRC-IEU | 462933 | NA | 43.854 | 0.0013 |
| Medication code: meloxicam | European | MRC-IEU | 462933 | NA | 20.6557 | 0.0003 |
| Medication code: naproxen | European | MRC-IEU | 462933 | NA | 20.8016 | 0.0004 |
| Number of medications taken | European | Neale Lab | 337159 | NA | 41.0722 | 0.0034 |
| **Disease-related factors** | | | | | | |
| IBS | European | NA | 486601 | 34741163 | 31.6799 | 0.0003 |
| GERD | European | NA | 602604 | 34187846 | 38.7677 | 0.0048 |
| Constipation | European | MRC-IEU | 463010 | NA | 21.1291 | 0.0005 |
| H.polyri infection | European | NA | 4683 | NA | 23.3411 | 0.0868 |
| Hyperthyroidism | European | The ThyroidOmics Consortium | 51823 | 30367059 | 41.5769 | 0.0119 |
| Hypothyroidism | European | The ThyroidOmics Consortium | 53423 | 30367059 | 37.4213 | 0.0070 |
| **Metabolism-related factors** | | | | | | |
| Type 1 diabetes | European | NA | 29652 | NA | 115.3071 | 0.1766 |
| Type 2 diabetes | European | NA | 433540 | NA | 82.2172 | 0.0264 |
| Fasting insulin | European | MAGIC | 51750 | 22581228 | 36.1021 | 0.0028 |
| Fasting glucose | European | MAGIC | 58074 | 22581228 | 95.8481 | 0.0351 |
| BMI | European | GIANT | 449889 | 29273807 | 85.7859 | 0.0169 |
| Obesity | European | MRC-IEU | 463010 | NA | 59.5773 | 0.0003 |
| Hypertension | European | NA | 484598 | 33959723 | 69.1705 | 0.0318 |
| Systolic blood pressure | European | NA | 810865 | 33230300 | 89.2834 | 0.0220 |
| NAFLD | European | NA | 778614 | 34841290 | 90.8944 | 0.0005 |
| **Sociological factors** | | | | | | |
| Educational attainment | European | NA | 461457 | 29892013 | 45.1189 | 0.0195 |
| Heavy physical work | European | MRC-IEU | 263615 | NA | 36.8653 | 0.0031 |
| Unpaid or voluntary work | European | MRC-IEU | 461242 | NA | 38.8940 | 0.0001 |
| Inability to work | European | UK Biobank | 461242 | NA | 31.6629 | 0.0003 |
| **Lifestyle-related factors** | | | | | | |
| Butter-cooked | European | UK Biobank | 51427 | NA | 21.1089 | 0.0069 |
| Drinks per week | European | GSCAN | 535425 | 30643251 | 75.1978 | 0.0050 |
| Vegetable oil cooked | European | UK Biobank | 51427 | NA | 30.5131 | 00006 |
| Low-calorie diet | European | UK Biobank | 51427 | NA | 21.4980 | 0.0091 |
| Gluten-free diet | European | UK Biobank | 51427 | NA | 22.4677 | 0.0151 |
| Cigarettes per day | European | GSCAN | 26201 | 30643251 | 63.7090 | 0.0050 |
| Olive oil | European | UK Biobank | 51427 | NA | 21.8129 | 0.0080 |
| Lard-cooked | European | UK Biobank | 51427 | NA | 22.7542 | 0.0234 |
| Vegetable oil | European | UK Biobank | 51427 | NA | 34.0050 | 0.0092 |
| Light physical activity | European | MRC-IEU | 460376 | NA | 41.0997 | 0.0011 |
| Heavy physical activity | European | MRC-IEU | 460376 | NA | 35.2107 | 0.0013 |
| Poultry intake | European | MRC-IEU | 461900 | NA | 31.8951 | 0.0005 |

*(Continued)*

**Table 1.** (Continued)

| Exposure | Ethnicity | Consortium | Total population | PMID | F- statistics | $R^2$ |
|---|---|---|---|---|---|---|
| Vegetarian diet | European | UK Biobank | 51427 | NA | 21.0094 | 0.0117 |
| Lactose-free diet | European | UK Biobank | 51427 | NA | 30.9586 | 0.0006 |
| **Psychological factors** | | | | | | |
| Autism spectrum disorder | European | PGC | 46351 | NA | 34.3546 | 0.0015 |
| Bipolar disorder | European | PGC | 51710 | NA | 34.8949 | 0.0080 |
| Anxiety | European | PGC | 83566 | NA | 22.5924 | 0.0082 |
| Miserableness | European | Neale Lab | 331856 | NA | 36.6674 | 0.0031 |
| Mood swings | European | MRC-IEU | 451619 | NA | 41.2380 | 0.0050 |
| Depression | European | PGC | 500199 | 2970045 | 39.0036 | 0.0036 |
| ADHD | European | PGC | 55374 | NA | 34.6050 | 0.0062 |
| Sleep disorders | European | NA | 361194 | NA | 21.6518 | 0.0011 |
| **Others** | | | | | | |
| Overall health rating | European | MRC-IEU | 460844 | NA | 41.0361 | 0.0090 |
| C-reactive protein levels | European | NA | 575531 | 35459240 | 192.1198 | 0.0728 |

Abbreviations: IBS, Irritable bowel syndrome ADHD, Attention Deficit Hyperactivity Disorder; GERD, Gastroesophageal reflux disease; BMI, Body mass index; NAFLD, Nonalcoholic fatty liver disease; GIANT, Genetic Investigation of ANthropometric Traits; GSCAN, GWAS and Sequencing Consortium of Alcohol and Nicotine use; MAGIC, Meta-Analyses of Glucose and Insulin-related traits Consortium; PGC, Psychiatric Genomics Consortium; MRC-IEU, MRC Integrative Epidemiology Unit; GSCAN: GWAS and Sequencing Consortium of Alcohol and Nicotine use.

To avoid the bias arising from different ethnicities, we only selected and analyzed genetic variants from the population of European ancestry as the current data sources. We obtained significant genetic variants at genome-wide association (GWA) level ($p < 5 \times 10^{-8}$) and deleted single nucleotide polymorphisms (SNPs) which might have existing linkage disequilibrium. In consideration of few SNPs associated with FD at the level of $p < 5 \times 10^{-8}$, the genetic instruments were set as $p < 1 \times 10^{-5}$. Eventually, we only included those SNPs with a long physical distance of $\geq 1,0000$ kb and a low likelihood of linkage disequilibrium ($r^2 < 0.001$).

## GWAS summary statistics of FD

Most exposure factors were extracted from UK biobank or PGC, so we obtained GWAS summary statistics of FD from FinnGen to decrease overlap. This dataset included 8875 FD cases and 320387 controls. FD were defined based on the International Classification of Diseases (ICD10 K30). In the FinnGen dataset, FD was defined as "An uncomfortable, often painful feeling in the stomach, resulting from impaired digestion. Symptoms include burning stomach pain, bloating, heartburn, nausea, and vomiting."

The current study accords with the Transparency and Openness Promotion guidelines. GWAS summary-level data supporting our findings are available in the public websites. All related studies were approved by relevant ethical boards. Informed consents were also obtained from all participants.

## Statistical analysis

Association intensities of genetic instruments for presumptive risk factors were quantified using F-statistic. F statistics ($F = beta^2 \div se^2$) were measured for each SNP and a general F statistic was measured for all SNPs. SNPs with F-statistic $> 10$ were considered as significant instrumental variables (Table 1) [18]. $R^2$ measures the proportion of the exposure's variability

explained by each SNP, and the total $R^2$ represents the extent to which instrumental variables explain exposure. MRnd was performed to calculate the statistical power for MR [19].

We presented results using a variety of sensitivity analyses that enabled the estimations to be accurate even when horizontal pleiotropy exists. The random effects IVW method was employed as the primary analytical approach to assess the relationships between genetic variants of risk factors with risks of FD [20]. Then sensitivity analyses were performed to evaluate the consistency of the findings using the weighted median model, MR-Egger and MR-PRESSO methods. The weighted median model allowed for unbiased causal estimates If more than 50% of those selected SNPs were valid [21]. MR Egger estimator was employed to obtain valid causal estimates even under the presence of horizontal pleiotropy [22]. The MR-PRESSO approach was used to identify pleiotropic outliers in summary-level multi-instrumental MR analysis, and to obtain causal effect estimates using the IVW model after removing these pleiotropic outliers [23].

Multivariate MR was also utilized as a supplementary strategy to evaluate the influence of significant exposure factors in the univariate MR model on the FD. A Leave-one-out sensitivity test was used to test the impact of outlying and pleiotropic genetic variants on causal estimates. Cochrane's Q statistics and MR-PRESSO were utilized to assess the heterogeneity and pleiotropy of single SNPs. A p value of $< 9.4 \times 10^{-4}$ (0.05/53) was considered as significant after Bonferroni correction. P values between $9.4 \times 10^{-4}$ and 0.05 were regarded to be indicative of potential associations. All analyses were conducted using statistical software R4.2.2 (R Foundation for Statistical Computing, Vienna, Austria), with "TwoSampleMR"0.5.6, "MR-PRESSO"1.0, and "MendelianRandomization"0.7.0 packages for processing and coordinating exposure and outcome data.

## Results

### Baseline characteristics

53 modifiable risk factors were included to assess possible causal relationships and categorized into eight categories. The values of F-statistics for the considered traits were greater than 10, indicating no potential instrument bias (Table 1). The number of SNPs ranged from 1 to 235 (Table 2). The following variants, including antidepressants, meloxicam, naproxen, gluten-free diet, HP infection, constipation, anxiety, sleep disorders, low-calorie diet, vegetarian diet, olive oil-cooked, butter-cooked and lard-cooked, were included using the selection criteria of $p < 1 \times 10^{-5}$ at GWA level. The remaining variants were included using the selection criteria of $p < 5 \times 10^{-8}$ (S2 Table in S2 File).

### Significant factors for the risks of FD

Genetically predicted GERD (OR 1.320, 95% CI 1.153 to 1.511, p = 0.0000567), less years of education (OR 0.926, 95% CI 0.894 to 0.958, p = 0.0000133) and depression (OR 1.515, 95% CI 1.231 to 1.865, p = 0.0000885) were found to be significantly associated with an increased risk of FD using the IVW method. Only genetically predicted of depression was significantly associated with an higher risk of FD using the WM method (OR 0.951, 95% CI 0.907 to 0.997, p = 0.039) (Figs 2, 3 and Table 2). We observed possible pleiotropy for depression ($P_{pleiotropy}$ = 0.0018) (S2 Table in S2 File). Thus, we conducted MR-PRESSO analysis after removing outliers, and the relationship remained stable in the corrected results (OR 1.591, 95% CI 1.313 to 1.928, p = 0.0000224). This result strongly suggested that depression was a significant predictor of an increased risk of FD. The estimated causal effects of each SNP associated with those factors on FD were also presented in the scatterplot (S1 Fig). The sensitivity analyses using leave-one-out plots illustrated the stability of the results (S2 Fig).

**Table 2. Possible risk factors for functional dyspepsia included in FinnGen consortium.**

| | SNPs | IVW OR (95% CI) | P value | SNPs | WM OR (95% CI) | P | SNPs | MR-Egger OR (95% CI) | P | SNPs | MR-PRESSO OR (95% CI) | P | Power |
|---|---|---|---|---|---|---|---|---|---|---|---|---|---|
| **Hormone-related factors** | | | | | | | | | | | | | |
| Testosterone levels | 94 | 0.953 (0.732 to 1.242) | 0.722 | 94 | 1.119 (0.711 to 1.762) | 0.626 | 94 | 0.847 (0.518 to 1.384) | 0.510 | 94 | NA | NA | 0.005 |
| Estradiol levels | 2 | 0.798 (0.588 to 1.082) | 0.146 | NA | NA | NA | NA | NA | NA | 2 | NA | NA | 0.008 |
| **Medication-related factors** | | | | | | | | | | | | | |
| Antihypertensives | 7 | 0.996 (0.850 to 1.166) | 0.956 | 7 | 0.977 (0.813 to 1.173) | 0.802 | 7 | 0.862 (0.423 to 1.756) | 0.699 | 7 | NA | NA | 0.024 |
| Antidepressants | 14 | 0.996 (0.894 to 1.108) | 0.936 | 14 | 0.976 (0.851 to ) | 0.729 | 14 | 0.862 (0.545 to 1.364) | 0.538 | 14 | NA | NA | 0.024 |
| Aspirin | 12 | 3.796 (0.492 to 29.307) | 0.201 | 12 | 4.846 (0.352 to 66.807) | 0.238 | 12 | 3.533 (0.019 to 655.073) | 0.646 | 12 | NA | NA | 0.989 |
| Ibuprofen | 3 | **0.005 (0.000 to 0.384)** | **0.017** | 3 | **0.007 (0.000 to 0.762)** | **0.038** | 3 | 104.165 (0.000 to 26544830302404.785) | 0.788 | 3 | NA | NA | 6.01e-24 |
| Paracetamol | 14 | 1.527 (0.293 to 7.959) | 0.615 | 14 | 0.659 (0.072 to 6.026) | 0.712 | 14 | 0.001 (0.000 to 1.380) | 0.086 | 14 | NA | NA | 0.299 |
| Meloxicam | 6 | 1567327.138 (0.000 to 8687555257578264.000) | 0.258 | 6 | 676756497.415 (0.000 to 3724410452644917674240.000) | 0.208 | 6 | $2.748\times10^{68}$ (0.000 to Inf) | 0.181 | 6 | NA | NA | 1 |
| Naproxen | 10 | **0.000 (0.000 to 0.085)** | **0.018** | 10 | 0.001 (0.000 to 6443.871) | 0.368 | 10 | 0.000 (0.000 to $2.272\times10^{31}$) | 0.696 | 10 | NA | NA | 6.15e-200 |
| Medications taken | 19 | **2.952 (1.550 to 5.622)** | **0.00099** | 19 | **2.693 (1.115 to 6.505)** | **0.028** | 19 | 0.855 (0.060 to 12.112) | 0.909 | 19 | NA | NA | 0.996 |
| **Disease-related factors** | | | | | | | | | | | | | |
| IBS | 4 | 1.455 (0.998 to 2.121) | 0.051 | 4 | 1.450 (0.916 to 2.296) | 0.113 | 4 | 0.124 (0.001 to 10.337) | 0.452 | 4 | NA | NA | 0.081 |
| GERD | 75 | **1.320 (1.153 to 1.511)** | **5.67E-05** | 75 | 1.107 (0.923 to 1.328) | 2.75E-01 | 75 | 0.957 (0.431 to 2.123) | 9.14E-01 | 75 | NA | NA | 0.431 |
| Constipation | 10 | 0.005 (0.000 to 736.178) | 0.382 | 10 | 0.000 (0.000 to 120.668) | 0.168 | 10 | 0.021 (0.000 to $3.133\times10^{39}$) | 0.933 | 10 | NA | NA | 3.25e-36 |
| H.polyri infection | 19 | 1.013 (0.944 to 1.087) | 0.723 | 19 | 1.028 (0.945 to 1.119) | 0.517 | 19 | 1.098 (0.930 to 1.296) | 0.288 | 19 | NA | NA | 0.054 |
| Hyperthyroidism | 15 | 1.034 (0.992 to 1.077) | 0.113 | 15 | 1.048 (0.994 to 1.104) | 0.084 | 15 | 1.061 (0.950 to 1.185) | 0.311 | 15 | NA | NA | 0.052 |
| Hypothyroidism | 10 | 0.986 (0.928 to 1.048) | 0.660 | 10 | 1.015 (0.932 to 1.104) | 0.738 | 10 | 1.026 (0.904 to 1.165) | 0.698 | 10 | NA | NA | 0.019 |
| **Metabolism-related factors** | | | | | | | | | | | | | |
| Type 1 diabetes | 35 | **1.032 (1.004 to 1.061)** | **0.027** | 35 | 1.025 (0.983 to 1.068) | 0.250 | 35 | 1.019 (0.970 to 1.070) | 0.463 | 35 | NA | NA | 0.229 |
| Type 2 diabetes | 143 | **0.956 (0.914 to 0.999)** | **0.043** | 143 | 0.938 (0.873 to 1.007) | 0.079 | 143 | 0.973 (0.886 to 1.067) | 0.559 | 143 | NA | NA | 0.004 |
| Fasting insulin | 4 | 1.101 (0.536 to 2.261) | 0.794 | 4 | 0.937 (0.392 to 2.238) | 0.883 | 4 | 0.073 (0.001 to 4.192) | 0.333 | 4 | NA | NA | 0.068 |
| Fasting glucose | 22 | 0.977 (0.797 to 1.198) | 0.825 | 22 | 1.025 (0.769 to 1.365) | 0.868 | 22 | 0.888 (0.583 to 1.352) | 0.585 | 22 | NA | NA | 0.009 |
| BMI | 90 | 0.872 (0.747 to 1.017) | 0.081 | 90 | 0.750 (0.577 to 0.974) | 0.031 | 90 | 0.900 (0.650 to 1.247) | 0.528 | 90 | NA | NA | 0.0001 |
| Obesity | 2 | 0.001 (0.000 to 157841.194) | 0.486 | NA | NA | NA | NA | NA | NA | 2 | NA | NA | 2.31e-32 |
| Hypertension | 230 | 1.107 (0.798 to 1.537) | 0.542 | 230 | 0.966 (0.611 to 1.526) | 0.881 | 230 | **2.423 (1.045 to 5.619)** | **0.040** | 230 | NA | NA | 0.394 |
| Systolic blood pressure | 204 | **1.188 (1.005 to 1.405)** | **0.043** | 204 | 1.189 (0.932 to 1.517) | 0.163 | 204 | 1.351 (0.908 to 2.012) | 0.140 | 204 | NA | NA | 0.661 |

(*Continued*)

**Table 2.** (Continued)

| | SNPs | IVW | | SNPs | WM | | SNPs | MR-Egger | | SNPs | MR-PRESSO | | Power |
|---|---|---|---|---|---|---|---|---|---|---|---|---|---|
| | | OR (95% CI) | P value | | OR (95% CI) | P value | | OR (95% CI) | P | | OR (95% CI) | P | |
| NAFLD | 4 | 0.965 (0.833 to 1.117) | 0.632 | 4 | 0.915 (0.815 to 1.026) | 0.127 | 4 | 0.886 (0.581 to 1.352) | 0.631 | 4 | NA | NA | 0.021 |
| **Sociological factors** | | | | | | | | | | | | | |
| Educational attainment | 203 | **0.926 (0.894 to 0.958)** | **1.33e-05** | 203 | **0.951 (0.907 to 0.997)** | **0.039** | 203 | 0.894 (0.769 to 1.040) | 0.147 | 203 | NA | NA | 0.002 |
| Heavy physical work | 22 | 1.633 (0.909 to 2.933) | 0.101 | 22 | 1.831 (0.913 to 3.672) | 0.089 | 22 | 0.273 (0.013 to 5.972) | 0.419 | 22 | NA | NA | 0.713 |
| Unpaid or voluntary work | NA | NA | NA | NA | NA | NA | NA | NA | NA | 1 | NA | NA | 0.0004 |
| Inability to work | 5 | 0.420 (0.000 to 433.824) | 0.807 | 5 | 0.044 (0.000 to 102.302) | 0.430 | 5 | 0.000 (0.000 to 92.388) | 0.213 | 5 | NA | NA | 0.0003 |
| **Lifestyle-related factors** | | | | | | | | | | | | | |
| Butter-cooked | 17 | 0.659 (0.309 to 1.402) | 0.279 | 17 | 0.495 (0.170 to 1.436) | 0.196 | 17 | 1.208 (0.314 to 4.649) | 0.787 | 17 | NA | NA | 1.04e-07 |
| Drinks per week | 36 | **1.566 (1.071 to 2.289)** | **0.021** | 36 | 1.472 (0.841 to 2.576) | 0.175 | 36 | 1.755 (0.714 to 4.314) | 0.229 | 36 | NA | NA | 0.839 |
| Vegetable oil cooked | NA | NA | NA | NA | NA | NA | NA | NA | NA | 1 | NA | NA | 1 |
| Low-calorie diet | 22 | 1.038 (0.522 to 2.063) | 0.915 | 22 | 1.247 (0.451 to 3.448) | 0.671 | 22 | 2.733 (0.556 to 13.438) | 0.230 | 22 | NA | NA | 0.052 |
| Gluten-free diet | 35 | 0.465 (0.156 to 1.387) | 0.170 | 35 | 0.713 (0.148 to 3.428) | 0.672 | 35 | 0.361 (0.048 to 2.700) | 0.328 | 35 | NA | NA | 5.04e-27 |
| Cigarettes per day | 21 | 0.983 (0.804 to 1.200) | 0.863 | 21 | 0.948 (0.749 to 1.200) | 0.658 | 21 | 0.795 (0.492 to 1.285) | 0.361 | 21 | NA | NA | 0.019 |
| Olive oil | 19 | 0.665 (0.401 to 1.104) | 0.115 | 19 | 0.553 (0.274 to 1.116) | 0.098 | 19 | 0.604 (0.211 to 1.735) | 0.363 | 19 | NA | NA | 4.45e-08 |
| Lard-cooked | 54 | 0.854 (0.287 to 2.537) | 0.776 | 54 | 1.132 (0.218 to 5.864) | 0.883 | 54 | 0.419 (0.071 to 2.461) | 0.340 | 54 | NA | NA | 1.29e-05 |
| Vegetable oil | 14 | 2.797 (0.611 to 12.808) | 0.185 | 14 | 2.742 (0.620 to 12.128) | 0.184 | 14 | 4.050 (0.309 to 53.124) | 0.308 | 14 | NA | NA | 1 |
| Light physical activity | 12 | 0.164 (0.009 to 2.872) | 0.216 | 12 | 0.836 (0.099 to 7.085) | 0.869 | 12 | 234.689 (0.004 to 15711271.759) | 0.358 | 10 | 0.152 (0.024 to 0.949) | 0.075 | 4.20e-14 |
| Heavy physical activity | 17 | 0.295 (0.051 to 1.698) | 0.171 | 17 | 0.573 (0.078 to 4.197) | 0.584 | 17 | 15.753 (0.000 to 1096765.625) | 0.635 | 17 | NA | NA | 7.12e-10 |
| Poultry intake | 7 | **4.882 (1.238 to 19.250)** | **0.024** | 7 | **5.141 (1.103 to 23.961)** | **0.037** | 7 | 13.284 (0.000 to 539827441742998339604) | 0.915 | 7 | NA | NA | 0.900 |
| Vegetarian diet | 29 | **0.297 (0.116 to 0.760)** | **0.011** | 29 | **0.190 (0.050 to 0.720)** | **0.015** | 29 | **0.124 (0.026 to 0.590)** | **0.014** | 29 | NA | NA | 6.76e-46 |
| Lactose-free diet | NA | NA | NA | NA | NA | NA | NA | NA | NA | 1 | NA | NA | 1 |
| **Psychological factors** | | | | | | | | | | | | | |
| Autism spectrum disorder | 2 | 1.072 (0.822 to 1.399) | 0.607 | 2 | NA | NA | 2 | NA | NA | 2 | NA | NA | 0.044 |
| Bipolar disorder | 12 | **1.156 (1.006 to 1.329)** | **0.041** | 12 | **1.179 (1.002 to 1.388)** | **0.047** | 12 | 0.753 (0.327 to 1.733) | 0.520 | 12 | NA | NA | 0.226 |

*(Continued)*

**Table 2.** (Continued)

| | SNPs | IVW OR (95% CI) | P value | SNPs | WM OR (95% CI) | P | SNPs | MR-Egger OR (95% CI) | P | SNPs | MR-PRESSO OR (95% CI) | P | Power |
|---|---|---|---|---|---|---|---|---|---|---|---|---|---|
| Anxiety | 41 | 0.949 (0.882 to 1.021) | 0.163 | 41 | 0.923 (0.836 to 1.019) | 0.111 | 41 | 1.195 (0.614 to 2.328) | 0.603 | 41 | NA | NA | 0.008 |
| Miserableness | 28 | 0.478 (0.218 to 1.049) | 0.066 | 28 | 0.564 (0.185 to 1.723) | 0.315 | 28 | 28.971 (1.191 to 704.781) | 0.049 | 28 | NA | NA | 4.09e-09 |
| Mood swings | 55 | 2.308 (1.113 to 4.783) | 0.025 | 55 | 2.464 (0.943 to 6.442) | 0.066 | 55 | 3.913 (0.048 to 317.217) | 0.546 | 55 | NA | NA | 1.000 |
| Depression | 46 | 1.515 (1.231 to 1.865) | 8.88e-05 | 46 | 1.440 (1.110 to 1.868) | 6.08e-03 | 46 | 0.236 (0.078 to 0.721) | 0.015 | 45 | 1.591 (1.313 to 1.928) | 2.24e-05 | 0.636 |
| ADHD | 10 | 0.955 (0.846 to 1.078) | 0.457 | 10 | 0.984 (0.848 to 1.141) | 0.832 | 10 | 1.166 (0.706 to 1.926) | 0.565 | 10 | NA | NA | 0.011 |
| Sleep disorders | 19 | 0.001 (1.17E-06 to 0.395) | 0.025 | 19 | 0.0001 (9.64E-09 to 2.00) | 0.069 | 19 | 0.008 (1.39E-07 to 413.207) | 0.392 | 19 | NA | NA | 1 |
| **Others** | | | | | | | | | | | | | |
| Overall health rating | 102 | 1.678 (1.175 to 2.396) | 0.004 | 102 | 1.421 (0.882 to 2.290) | 0.149 | 102 | 0.661 (0.117 to 3.722) | 0.639 | 102 | NA | NA | 0.995 |
| C-reactive protein levels | 235 | 0.985 (0.916 to 1.059) | 0.689 | 235 | 1.019 (0.895 to 1.161) | 0.771 | 235 | 1.000 (0.905 to 1.105) | 0.996 | 235 | NA | NA | 0.010 |

Abbreviations: IBS, Irritable bowel syndrome; GERD, Gastroesophageal reflux disease; ADHD, Attention Deficit Hyperactivity Disorder; BMI, Body mass index; NAFLD, Nonalcoholic fatty liver disease; SNPs, single nucleotide polymorphisms; IVW, Inverse variance weighted; WM, weighted median; MR-PRESSO, the MR pleiotropy residual sum and outlier. "NA" in the MR-PRESSO form indicated that the instrumental variables for this exposure did not have these pleiotropic outliers. The MR-PRESSO approach and the IVW method had the same result.

## Suggestive factors for the risks of FD

Genetically predicted type 1 diabetes (OR 1.032, 95% CI 1.004 to 1.061, p = 0.027), elevated SBP (OR 1.188, 95% CI 1.005 to 1.405, p = 0.043), higher numbers of medications taken (OR 2.952, 95% CI 1.550 to 5.622; p = 0.00099), bipolar disorder (OR 1.156 95% CI 1.006 to 1.329; p = 0.041), mood swings (OR 2.308 95% 1.113 to 4.783; P = 0.025), alcohol consumption frequency (OR 1.566, 95% CI 1.071 to 2.289;p = 0.021), Poultry intake (OR 4.882, 95% CI 1.238 to 19.250, p = 0.024) and lower overall health rating (OR 1.678, 95% CI 1.175 to 2.396, p = 0.004) were suggestively associated with an increased risk of FD using IVW method (Figs 2, 3 and Table 2).

Genetically predicted type 2 diabetes (OR 0.938 95% CI 0.873 to 1.007, p = 0.043), predicted taking ibuprofen (OR 0.005, 95% CI 0.000 to 0.384; p = 0.017) and taking naproxen (OR 0.000, 95% CI 0.000 to 0.085; p = 0.018), being vegetarian (OR 0.297, 95% CI 0.116 to 0.760; p = 0.011) and sleep disorder (OR 0.001, 95% CI 1.17E-06 to 0.395; p = 0.025) were observed to be suggestively lower risk factors for FD using IVW method (Figs 2, 3 and Table 2).

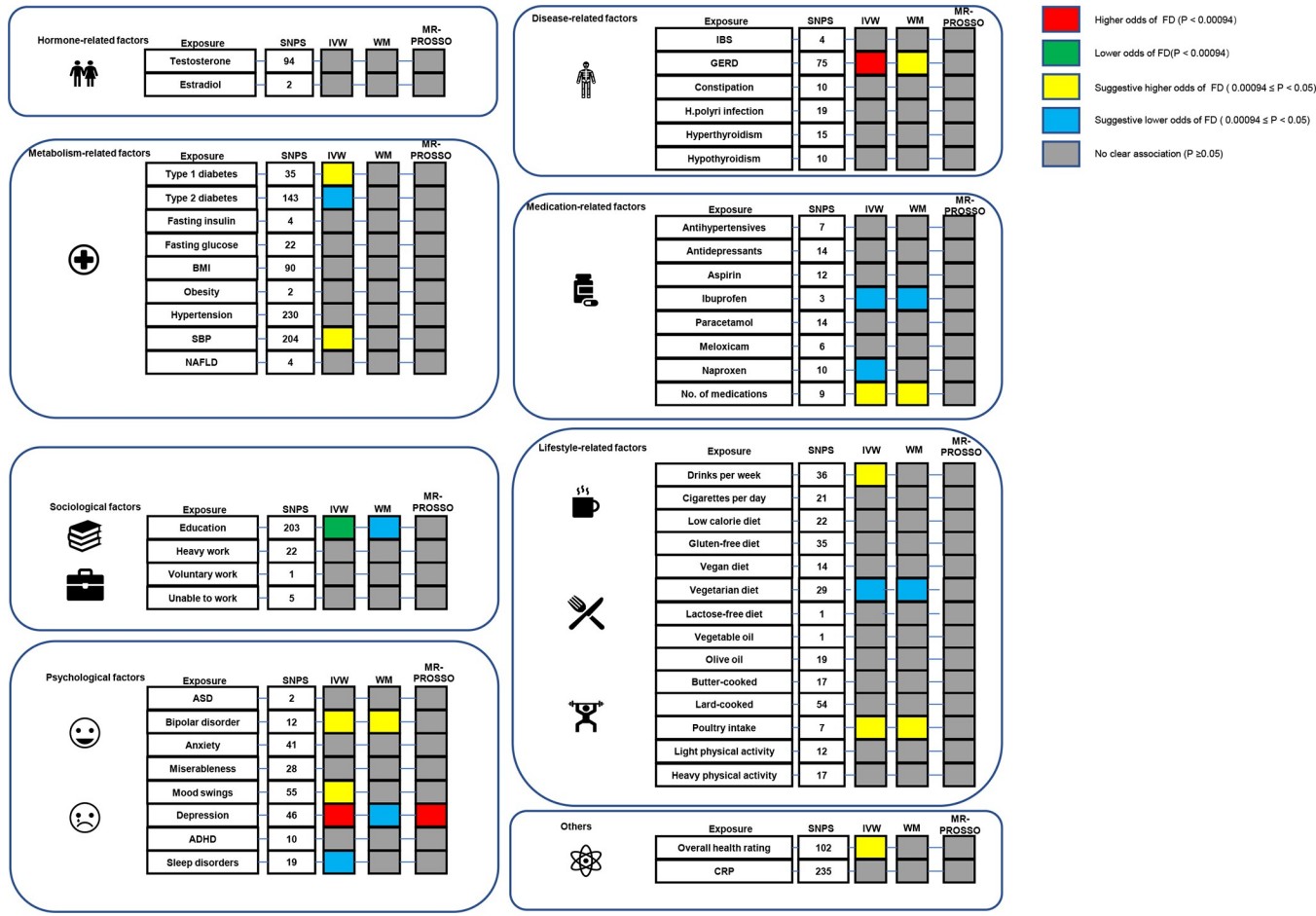

**Fig 2. The analyses of possible risk factors and FD in FinnGen datasets.** Abbreviations: BMI, Body mass index; SBP, Systolic blood pressure; NAFLD, Non-alcoholic fatty liver disease; ASD, Autism spectrum disorder; ADHD, Attention deficit hyperactivity disorder; IBS, Irritable bowel syndrome; GERD, Gastroesophageal reflux disease; CRP, C-reactive protein; IVW, Inverse variance weighted; WM, weighted median; MR-PRESSO, the MR pleiotropy residual sum and outlier.

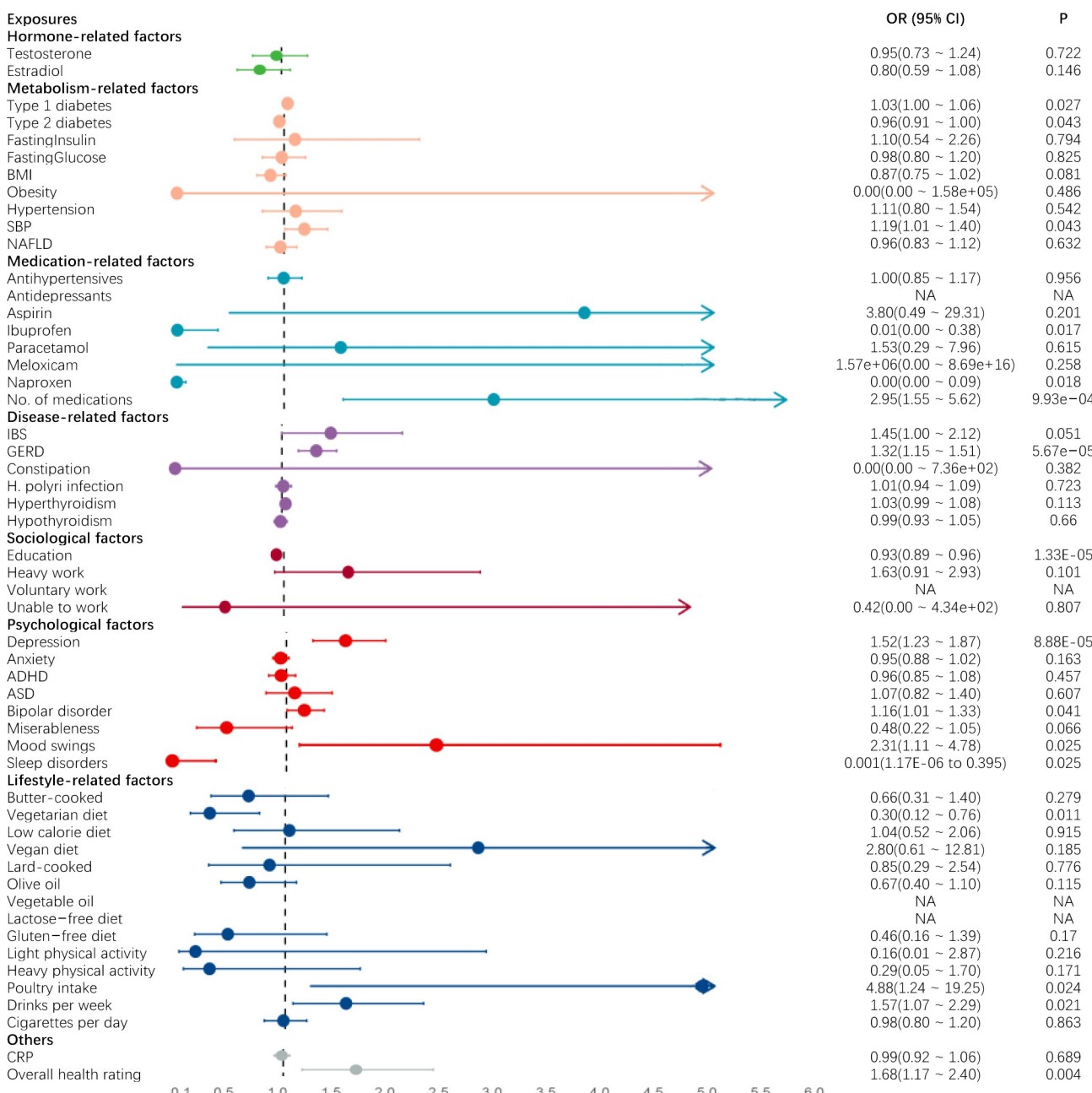

**Fig 3. The forest plots depicting the factors for the increased risk of FD.** Abbreviations: BMI, Body mass index; SBP, Systolic blood pressure; NAFLD, Non-alcoholic fatty liver disease; ASD, Autism spectrum disorder; ADHD, Attention deficit hyperactivity disorder; IBS, Irritable bowel syndrome; GERD, Gastroesophageal reflux disease; CRP, C-reactive protein.

The estimated causal effects of each SNP associated with those factors on FD were also presented in the scatterplot (S3 Fig). The sensitivity analyses using leave-one-out plots illustrated the stability of the results (S4.1–S4.4 Fig).

## Non-significant factors for the risks of FD

No significant causal associations were recorded between testosterone, Estradiol, fasting Insulin, fasting glucose, BMI, obesity, hypertension, NAFLD, IBS, constipation, HP infection,

| Exposure | Outcome | IVW | OR (95% CI) | P |
|---|---|---|---|---|
| Depression | FD | | 1.632 (1.175 to 2.266) | 0.003 |
| GERD | FD | | 0.854 (0.624 to 1.169) | 0.325 |
| Education | FD | | 0.907(0.839 to 0.980) | 0.014 |

*MVMR of Depression, Education, GERD and FD*

**Fig 4. Multivariate MR analyses of risk factors for FD.** Abbreviations: GERD, Gastroesophageal reflux disease.

hyperthyroidism, hypothyroidism, antihypertensives, antidepressants, aspirin, paracetamol, meloxicam, miserableness, heavy physical work, voluntary work, inability to work, ASD, anxiety, ADHD, various cooking oils, multiple dietary patterns, smoking frequency, physical exercise intensity, CRP, and FD (Figs 2, 3 and Table 2).

## Multivariate MR analyses of risk factors for FD

Multivariate MR analyses demonstrated that genetically predicted depression and educational attainment were significant causal factors for FD. The ORs (95% CI) were 1.632(1.175 to 2.266; p = 0.003) and 0.907(0.839 to 0.980; p = 0.014), respectively (Fig 4). GERD was not a causal factor for FD (OR 0.854, 95% CI 0.624 to 1.169; p = 0.325) (S3 Table in S2 File).

## Discussion

Although some potential risk factors for FD were identified in some previous observational studies and also confirmed by meta-analyses (shown in S1 Table in S2 File), potential confounding and reverse causality were innegligible issues in those studies. Therefore, we conducted a comprehensive MR study to evaluate causal relationships between 53 potential risk factors and FD. Our findings indicated that genetic predisposition to depression is a strong predictor of an increased risk of FD and genetic predisposition to longer years of education is a strong predictor of a decreased risk of FD. Whereas, the other risk factors, including obesity, HP infection, taking NSAIDs and anxiety, which were proven to be significantly linked to FD in previous observational studies, failed to demonstrate to their associations with FD at genetic level. To our knowledge, the current study was the first study to demonstrate the causal effects of depression and education attainment on FD. The duration of MR exposure to those risk factors is lifelong and the effects of the factors on FD may be profound. Therefore, the identification, follow-up and psychotherapy for individuals with depression were of vital importance for superior prevention and treatment of FD. In addition, our study also revealed the lifelong effect of education status on FD and reinforced the importance of education for FD.

FD was well-recognized as a disorder closely linked to psychological disorders. The communications between the brain and gut through enteric neurohumoral systems played important roles in the pathogenesis of FD [24]. Some psychological factors, including depression, anxiety, somatization, alexithymia and sleep disorder, were extensively investigated to identify their contributions to FD [25–27]. Among those factors, depression and anxiety were most frequently explored and demonstrated to be significant psychological factors associated with FD [28]. However, only depression was found to be a causal factor for FD at the genetic predisposition level in the current study. The finding was consistent with a 12-year prospective population-based study [10], in which only higher levels of depression at baseline were predictive for FD at follow-up. Moreover, we also did reverse MR and found FD was not a causal risk factor for depression (S5 Fig). Therefore, it demonstrated that could be a causal precursor for

subsequent FD, but this is not vice versa. This finding that genetic prediction to depression was significantly related to FD reinforces the aetiological importance of depression in FD. Dysfunction of brain-to-gut signals induced by depression has been demonstrated to involve in the generation of dyspepsia symptoms, such as abdominal pain and meal-related discomfort [29–31]. Therefore, the early identification of depression and active psychosocial interventions in patients with depression would contribute to preventing subsequent occurrences of persistent FD. In addition, a recent meta-analysis also revealed the beneficial effects of anti-depressant therapies on FD [32]. Furthermore, there are also requirements for psychological health education and emotional protection in the early stages of age to prevent an onset of FD [7].

Previous studies demonstrated that individuals with lower educational attainment may be at a higher risk of developing dyspepsia [14, 33, 34]. Our current MR analysis also confirmed the finding. The relationship may still be unclear until now. Individuals with lower educational attainment may have poorer knowledge and lower socioeconomic status and are prone to suffer from depression, which may contribute to the development of functional dyspepsia [35]. However, it is important to note that educational attainment is only one aspect of an individual's overall socioeconomic status and does not fully reflect their level of health literacy. Health literacy refers to an individual's ability to access, understand, and apply health information and make appropriate health decisions [36]. Functional dyspepsia is a condition strongly influenced by psychological factors, and individuals with lower health literacy may be more susceptible to developing functional dyspepsia due to their poorer ability to cope with stress [35].

FD was often found to co-occur with other functional gastrointestinal disorders such as GERD or IBS [37, 38]. However, to our best knowledge, there were no longitudinal studies focusing on whether GERD or IBS was the casual factor of FD. In our study, we included the two disease-related factors, as well as constipation, HP infection, hyperthyroidism and hypothyroidism, to explore the causal relationship between the above factors and FD. We found that genetically predicted GERD was significantly associated with FD in univariate MR analysis. The relationship between GERD and FD is complex and multifaceted. It is possible that FD and GERD share some common etiological factors such as delayed gastric emptying, impaired esophageal motility, and visceral hypersensitivity [39]. However, multivariate MR indicated that GERD was not a causal risk factor for FD independent of depression and education attainment, suggesting this association could be influenced by the existence of depression and education.

Previous studies reported individuals with alcohol intake and diabetes are also accompanied by dyspeptic symptoms, and excessive alcohol intake and diabetes were commonly regarded as causes of organic dyspepsia [40, 41]. In our study, genetically predicted alcohol consumption and type 1 diabetes are suggestive of higher odds for FD. However, prior studies have failed to verify a causal relationship between excessive alcohol consumption or diabetes with FD. In addition, poultry intake, elevated systolic blood pressure and lower overall health rating, which were not included as investigated factors in previous studies, also showed suggestive higher odds for FD. Both suggestive increased risks for FD and unclear rationales make the findings of little clinical significance. Therefore, further well-designed cohorts were required to be designed to clarify the potential correlations. The other modifiable risk factors, such as high-fat diet, usage of NSAIDs and obesity, could be predictive risk factors for FD in observational studies [42–44]. Nonetheless, our MR analysis did not confirm their genetically significant relationships with FD. The findings suggested that those modifiable factors impact FD risk as outcomes of shared risk profiles.

There were several limitations in the study. Firstly, horizontal pleiotropy was an issue that cannot be ignored in MR studies [20, 23]. This was also the case in our MR analysis. Various sensitivity analyses, including the MR-PRESSO test, intercept test, heterogeneity test, and

leave-one-out analysis were performed reduce bias. Stable results were observed after above. Secondly, the detailed subtypes of FD were not recorded in the current gene datasets and therefore not included in this study. Clauwaert et al revealed a significant association between mental disorders and postprandial distress syndrome (PDS), not epigastric pain syndrome (EPS) [45]. Thus, it was interesting to investigate the relationship between psychological disorders and subtypes of FD in further studies. Thirdly, The definition of FD in the Finngen database is a boarder definition [46], which was not as exact as the well-recognized Rome IV criteria [24].Rome IV criteria was widely adopted for the purposes of diagnosis and treatment of functional gastrointestinal disorders, as well as scientific research in clinical trials. However, this criteria has not been included in any GWA datasets. Fouthly, the present study only consisted of individuals of European ancestry. Although population stratification bias was reduced in methology, the population restriction may limit the generalizability of the findings to other populations.

In conclusion, our comprehensive MR study demonstrated that depression and lower educational attainment were the causal factors for functional dyspepsia in the genetic levels. The early identification of depression and Improvement of educational attainment were essential for superior prevention and treatment of FD.

## Supporting information

**S1 Fig. The estimated causal effects of each SNP associated with significant factors on FD.**
(TIF)

**S2 Fig. Leave-one-out sensitivity test for significant factors.**
(TIF)

**S3 Fig. The estimated causal effects of each SNP associated with suggestive factors on FD.**
(TIF)

**S4 Fig.** S4.1 Fig. Leave-one-out sensitivity test for suggestive factor (systolic blood pressure). S4.2 Fig. Leave-one-out sensitivity test for suggestive factor (type 2 diabetes). S4.3 Fig. Leave-one-out sensitivity test for suggestive factor (overall health rating). S4.4 Fig. Leave-one-out sensitivity test for suggestive factors (type 1 diabetes, bipolar disorder, mood swings, poultry intake, etc).
(ZIP)

**S5 Fig. Reverse MR for FD on depression.**
(PNG)

**S1 File. STROBE-MR checklist of recommended items to address in reports of Mendelian randomization studies.**
(DOCX)

**S2 File. Supplementary tables providing necessary information (S1-S3 Tables).**
(DOCX)

## Author Contributions

**Formal analysis:** Zeyu Wang, Tangyi Liu.

**Software:** Tangyi Liu, Ze Yang.

**Supervision:** Hui Luo, Yanglin Pan.

**Writing – original draft:** Zeyu Wang, Dan Cao, Xiaoyu Kang.

**Writing – review & editing:** Xiaoyu Kang, Yanglin Pan.

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
