## [Decision Letter · Decision Letter 0]

2 Apr 2024

PONE-D-24-04501The associations between possible risk factors and functional dyspepsia: A comprehensive Mendelian randomization study.PLOS ONE

Dear Dr. Pan,

Thank you for submitting your manuscript to PLOS ONE. After careful consideration, we feel that it has merit but does not fully meet PLOS ONE’s publication criteria as it currently stands. Therefore, we invite you to submit a revised version of the manuscript that addresses the points raised during the review process.

We look forward to receiving your revised manuscript.

Kind regards,

Jorge Cervantes

Academic Editor

PLOS ONE

“This work supported in part by National Key R&D Program of China (2022YFC2505100) and the National Natural Science Foundation of China (81970557 and 82373117).”

3. PLOS requires an ORCID iD for the corresponding author in Editorial Manager on papers submitted after December 6th, 2016. Please ensure that you have an ORCID iD and that it is validated in Editorial Manager. To do this, go to ‘Update my Information’ (in the upper left-hand corner of the main menu), and click on the Fetch/Validate link next to the ORCID field. This will take you to the ORCID site and allow you to create a new iD or authenticate a pre-existing iD in Editorial Manager. Please see the following video for instructions on linking an ORCID iD to your Editorial Manager account: https://www.youtube.com/watch?v=_xcclfuvtxQ.

Additional Editor Comments:

Dear Dr. Pn,

You manuscript has undergone careful review, and some revisions are needed before it can be suitable for publication.

Reviewers' comments:

Reviewer's Responses to Questions

**Comments to the Author**

1. Is the manuscript technically sound, and do the data support the conclusions?

Reviewer #1: Yes

Reviewer #2: Yes

2. Has the statistical analysis been performed appropriately and rigorously? 

Reviewer #1: Yes

Reviewer #2: Yes

3. Have the authors made all data underlying the findings in their manuscript fully available?

Reviewer #1: Yes

Reviewer #2: Yes

4. Is the manuscript presented in an intelligible fashion and written in standard English?

Reviewer #1: Yes

Reviewer #2: No

5. Review Comments to the Author

Reviewer #1: I have carefully reviewed your manuscript titled "The associations between possible risk factors and functional dyspepsia: A comprehensive Mendelian randomization study." and would like to provide you with some constructive feedback to improve the quality and impact of your work. Overall, I find your study to be interesting and potentially valuable in understanding the causal relationship between human metabolites and functional dyspepsia.However, there are several areas that require attention.

1.supplementry figure 4：The text of the plot overlaps, so I recommend to draw it separately.

2.The duration of MR Exposure is lifelong, and the effect may be larger than that of clinical RCT studies.The results of MR Analysis do not indicate that the RCT study is necessarily valid.The conclusions should be interpreted with caution.

3.Figure1's MRII assumes that the location table notes should be accurate.Just next to this sentence"Genetic instruments are independent of confounding factors"

4.Please provide an explanation for the predominance of NA values in table 2 as a result of MR-PRESSO.

Overall, this paper is methodologically rigorous and meticulous, providing new insights into the causes of functional dyspepsia.

Reviewer #2: Overall, it is an interesting and relevant study (design / MR analyses is unique, considering the condition). Functional dyspepsia is a common and difficult to treat entity. Understanding the risk factors (other than psychological) is important to provide better therapies (treat the underlying factors). The conclusion is supported by the data - I greatly enjoyed the recommendations brought forth by the authors - it's pertinent in clinical practice. Recommend accepting for publication after making revisions (English / correcting grammar) - I made a few recommendations but there are additional sections that I did not mark (too many). I recommend an English editor for additional proofreading (this will make it easier on the reader and avoid distraction from the very interesting study / results).

Consider changing the title to include function dyspepsia first (the studied condition), and avoid the use of ‘possible’ in titles - consider ‘potential’ (i.e., The associations between functional dyspepsia and potential risk factors: a comprehensive Mendelian randomization study).

Under ‘Abstract / Background’ - change ‘identified many possible risk factors’ to ‘identified multiple potential risk factors.’ Change ‘between the factors and FD’ to ‘between these factors and FD.’

Under ‘Abstract / Methods’ - remove the first word ‘Totally’ - consider ‘A total of 53 possible.’ Change ‘obtained through comprehensive literature reviewing’ to ‘obtained through a comprehensive literature review.’ Don’t use the word ‘Totally’ in the manuscript - change to ‘A total of.’ Change ‘involved’ to ‘available’ (320387 controls were available for analysis).

Under ‘Abstract / Results’ - Change ‘was significantly associated with the risk for FD in univariate MR analyses’ to ‘were associated with risk for FD in univariate MR analyses.’ Change ‘alcohol drinking’ to ‘alcohol consumption.’ Change ‘diabetes type 1’ to ‘type 1 diabetes.’ 'Systolic blood pressure' requires additional details - I believe it should mention 'elevated' systolic blood pressure.’ Change ‘lower health ratings’ to ‘lower overall health rating.’

Under ‘Abstract / Conclusions’ - Consider re-wording it to ‘Our comprehensive MR study demonstrated that depression and lower educational attainment were causal factors for FD at the genetic level.’

Under ‘Introduction’ - Change ‘However, up to date, no studies performed MR analyses to explore the associations between modifiable risk factors and FD’ to ‘However, no study has been performed that uses MR analyses to explore the associations between modifiable risk factors and FD.’

Under ‘Methods / MR design’ - Change ‘published after 1989 Since Rome criteria was initially established’ to ‘published after 1989, when Rome criteria was initially established.’

There are multiple misspelled words in the ‘Discussion’ - at least ‘depression,’ ‘demonstrate.’

Under ‘Discussion’ - Re-word / grammatically correct the following ‘In our study, genetically predicted alcohol drinking and diabetes type 1 are suggested to have suggestive higher odds for FD. However, Up to date, no studies have verified causal relationships between excessive alcohol consumption, and diabetes with FD.’ Consider ‘In our study, genetically predicted alcohol consumption and type 1 diabetes are suggestive of higher odds for FD. However, prior studies have failed to verify a causal relationship between excessive alcohol consumption or diabetes with FD.’ Using the words ‘Up to date’ is not required - if needed, a sentence can be started with “To date,.”

6. PLOS authors have the option to publish the peer review history of their article (what does this mean?). If published, this will include your full peer review and any attached files.

Reviewer #1: No

Reviewer #2: No

---

## [Author Response · Author response to Decision Letter 0]

11 Apr 2024

Dear editor,

We thank the reviewers and editors for reviewing our Manuscript ID PONE-D-24-04501 entitled “The associations between functional dyspepsia and potential risk factors: A comprehensive Mendelian randomization study”. We have revised the manuscript according to the constructive suggestions. A point-by-point response is included below. 

In response to the comments of the editor and reviewers, additional information has been added to the revised manuscript. These are marked by yellow highlights in the version with tracked changes. The version with all changes accepted (clean version) are also provided. Deleted text no longer appears in the final clean version.

Reviewer #1

1.supplementry figure 4：The text of the plot overlaps, so I recommend to draw it separately.

Response: Thank you for your suggestion. The text of 3 plots overlapped in the supplementary figure 4. To make it clearer, we re-drew the three pictures and uploaded the three figures as separate supplementary materials. The new figures were presented as supplementary figures 4.1-4.4. 

2.The duration of MR Exposure is lifelong, and the effect may be larger than that of clinical RCT studies. The results of MR Analysis do not indicate that the RCT study is necessarily valid. The conclusions should be interpreted with caution.

Response: Thank you for your suggestion. Your comments make us understand the longer and larger effect of the MR analysis. Therefore, based on your suggestion, we added the following sentences as “To our knowledge, the current study was the first study to demonstrate the causal effects of depression and education attainment on FD. The duration of MR exposure to those risk factors is lifelong and the effects of the factors on FD may be profound. Therefore, the identification, follow-up and psychotherapy for individuals with depression were of vital importance for superior prevention and treatment of FD. In addition, our study also revealed the lifelong effect of education status on FD and reinforced the importance of education for FD.” in the first paragraph of Discussion Section with yellow highlighted. We hoped that the above discussions could make the audiences understand the clinical importance of our findings deeply.

3. Figure1's MRII assumes that the location table notes should be accurate. Just next to this sentence" Genetic instruments are independent of confounding factors".

Response: Thank you for your suggestion. According to your suggestion, we have placed the description of “MRII” next to the description of "Genetic instruments are independent of confounding factors" in Figure 1.

4.Please provide an explanation for the predominance of NA values in table 2 as a result of MR-PRESSO.

Response: Thank you for your question. To make it clearer, we added the following explanations as “The MR-PRESSO approach was used to identify pleiotropic outliers in summary-level multi-instrumental MR analysis, and to obtain causal effect estimates using the IVW model after removing these pleiotropic outliers.” in the sixth paragraph of the methods section with yellow highlighted. “NA” indicated that the instrumental variables for this exposure did not have these pleiotropic outliers. The MR-PRESSO approach and the IVW method had the same result. To make it clearer, we added the following contents as ““NA” in the MR-PRESSO form indicated that the instrumental variables for this exposure did not have these pleiotropic outliers. The MR-PRESSO approach and the IVW method had the same result.” in the Notes section below table 2 with yellow highlighted.

Reviewer #2

Consider changing the title to include function dyspepsia first (the studied condition), and avoid the use of ‘possible’ in titles - consider ‘potential’ (i.e., The associations between functional dyspepsia and potential risk factors: a comprehensive Mendelian randomization study).

Response: Thank you for your suggestion. According to your suggestion, we have modified this title as “The associations between functional dyspepsia and potential risk factors: a comprehensive Mendelian randomization study”. This new title will enable the audience to understand the aim of this article better. The modified contents were also marked with yellow highlighted in the updated version.

Under ‘Abstract / Background’ - change ‘identified many possible risk factors’ to ‘identified multiple potential risk factors.’ Change ‘between the factors and FD’ to ‘between these factors and FD.’

Response: Thank you for your suggestion. We have changed “identified many possible risk factors” to “identified multiple potential risk factors”. We have changed “between the factors and FD” to “between these factors and FD” in the Abstract Section. The modified contents were also marked with yellow highlighted.

Under ‘Abstract / Methods’ - remove the first word ‘Totally’ - consider ‘A total of 53 possible.’ Change ‘obtained through comprehensive literature reviewing’ to ‘obtained through a comprehensive literature review.’ Don’t use the word ‘Totally’ in the manuscript - change to ‘A total of.’ Change ‘involved’ to ‘available’ (320387 controls were available for analysis).

Response: Thank you for your suggestion. According to your suggestion, we have changed “Totally” to “A total of”, “obtained through comprehensive literature reviewing” to “obtained through a comprehensive literature review”, and “involved” to “available” in the Methods section of Abstract with yellow highlighted.

Under ‘Abstract / Results’ - Change ‘was significantly associated with the risk for FD in univariate MR analyses’ to ‘were associated with risk for FD in univariate MR analyses.’ Change ‘alcohol drinking’ to ‘alcohol consumption.’ Change ‘diabetes type 1’ to ‘type 1 diabetes.’ 'Systolic blood pressure' requires additional details - I believe it should mention 'elevated' systolic blood pressure.’ Change ‘lower health ratings’ to ‘lower overall health rating.’

Response：Thank you for your suggestion. According to your suggestion, we have changed “was significantly associated with the risk for FD in univariate MR analyses” to “were associated with risk for FD in univariate MR analyses”, “alcohol drinking” to “alcohol consumption”, “diabetes type 1” to “type 1 diabetes”, “Systolic blood pressure” to “elevated systolic blood pressure”, and “lower health ratings” to “lower overall health rating” in the Results section of Abstract with yellow highlighted.

Under ‘Abstract / Conclusions’ - Consider re-wording it to ‘Our comprehensive MR study demonstrated that depression and lower educational attainment were causal factors for FD at the genetic level.’

Response：Thank you for your suggestion. According to your suggestion, we have changed “Our comprehensive MR study demonstrated that depression and less years of education were the causal factors for FD at the genetic levels.” to “Our comprehensive MR study demonstrated that depression and lower educational attainment were causal factors for FD at the genetic level.” in the Conclusions section of Abstract with yellow highlighted.

Under ‘Introduction’ - Change ‘However, up to date, no studies performed MR analyses to explore the associations between modifiable risk factors and FD’ to ‘However, no study has been performed that uses MR analyses to explore the associations between modifiable risk factors and FD.’

Response：Thank you for your suggestion. According to your suggestion, we have changed” However, up to date, no studies performed MR analyses to explore the associations between modifiable risk factors and FD.” to “However, no study has been performed that uses MR analyses to explore the associations between modifiable risk factors and FD.” in the third paragraph of the Introduction section with yellow highlighted. 

Under ‘Methods / MR design’ - Change ‘published after 1989 Since Rome criteria was initially established’ to ‘published after 1989, when Rome criteria was initially established.’

Response：Thank you for your suggestion. According to your suggestion, we have changed” published after 1989 Since Rome criteria was initially established” to “published after 1989, when Rome criteria was initially established” in the MR design of Methods section with yellow highlighted.

There are multiple misspelled words in the ‘Discussion’ - at least ‘depression,’ ‘demonstrate.’

Response：Thank you for your suggestion. We have made corrections to the misspelled words in the first paragraph of the Discussion section with yellow highlighted. 

Under ‘Discussion’ - Re-word / grammatically correct the following ‘In our study, genetically predicted alcohol drinking and diabetes type 1 are suggested to have suggestive higher odds for FD. However, Up to date, no studies have verified causal relationships between excessive alcohol consumption, and diabetes with FD.’ Consider ‘In our study, genetically predicted alcohol consumption and type 1 diabetes are suggestive of higher odds for FD. However, prior studies have failed to verify a causal relationship between excessive alcohol consumption or diabetes with FD.’ Using the words ‘Up to date’ is not required - if needed, a sentence can be started with “To date,.”

Response: Thank you for your suggestion. According to your suggestion, we have changed” In our study, genetically predicted alcohol drinking and diabetes type 1 are suggested to have suggestive higher odds for FD. However, Up to date, no studies have verified causal relationships between excessive alcohol consumption, and diabetes with FD.” to “In our study, genetically predicted alcohol consumption and type 1 diabetes are suggestive of higher odds for FD. However, prior studies have failed to verify a causal relationship between excessive alcohol consumption or diabetes with FD.” in the fifth paragraph of the Discussion section with yellow highlighted. 

Sincerely,

Yanglin Pan

---

## [Editor Report · Decision Letter 1]

15 Apr 2024

The associations between functional dyspepsia and potential risk factors: A comprehensive Mendelian randomization study.

PONE-D-24-04501R1

Dear Dr. Pan,

We’re pleased to inform you that your manuscript has been judged scientifically suitable for publication and will be formally accepted for publication once it meets all outstanding technical requirements.

Kind regards,

Jorge Cervantes

Academic Editor

PLOS ONE
---

## [Editor Report · Acceptance letter]

26 Apr 2024

PONE-D-24-04501R1 

PLOS ONE

Dear Dr. Pan, 

I'm pleased to inform you that your manuscript has been deemed suitable for publication in PLOS ONE. Congratulations! Your manuscript is now being handed over to our production team.

Kind regards, 

on behalf of

Dr. Jorge Cervantes 

Academic Editor

PLOS ONE